# Physiological Potential of Seeds of *Handroanthus spongiosus* (Rizzini) S. Grose (Bignoniaceae) Determined by the Tetrazolium Test

**Jailton de Jesus Silva** [1], **Raquel Araujo Gomes** [1], **Maria Aparecida Rodrigues Ferreira** [1], **Claudineia Regina Pelacani** [1] and **Bárbara França Dantas** [2,*]

1    Departamento de Ciências Biológicas, Universidade Estadual de Feira de Santana, Avenida Transnordestina, s/n-Novo Horizonte, Feira de Santana 44036-900, BA, Brazil
2    Embrapa Semiárido, Rodovia BR-428, Km 152, Zona Rural, Petrolina 56302-970, PE, Brazil
*    Correspondence: barbara.dantas@embrapa.br; Tel.: +55-87-38663678

**Abstract:** Tetrazolium test (TZT) can quickly evaluate in detail the viability and vigor of seeds. This study aimed to determine the optimal conditions for conducting the TZT on seeds of *Handroanthus spongiosus*. For this purpose, seeds from three lots were pre-soaked in water for 16 h, followed by extraction of the tegument and immersion in tetrazolium salt solutions at different concentrations (0.01–0.1%), for increasing periods (1–4 h) and at 30 °C in the dark. The experimental design was completely randomized with a 4 × 4 factorial scheme with 25 seeds per repetition. We applied generalized linear models and the Tukey test for pairwise comparisons of the means at 5% probability. The viability/vigor results were compared with data obtained from the germination test at 25 °C using a subsample of seeds from the same lots. The time (1 h to 4 h) of immersion of the seeds in tetrazolium salt solutions did not cause a clear coloration difference. The seeds subjected to all treatment concentrations for 3 h presented average viability greater than 60%, with no difference in germination percentage. The TZT at 0.01% tetrazolium salt solution for 3 h was most efficient in assessing the viability of the *Handroanthus spongiosus* seeds.

**Keywords:** germination; forest seeds; vigor tests; sete-cascas

## 1. Introduction

Evaluation of the physiological potential of forest seeds is an important functional feature recommended soon after processing, before and after storage of seeds. Furthermore, evaluations of stored seeds' fitness allow us to distinguish lots with better quality and aid the choice of mother trees in the field with superior quality for seed harvest [1,2].

The germination test and tetrazolium test (TZT) are among the main methods used to assess the physiological potential of seeds [3,4]. The germination test is used most frequently for the official evaluation of the viability and vigor of seed lots in quality control programs [5]. However, the diagnosis of the seed quality by this test can take several days or even weeks, which can be too long in the case of recalcitrant seeds and seeds with low viability or longevity [6]. For example, the evaluation by the standard germination test of seeds of some species of the genus *Handroanthus* can take between 14 to 21 days [7]. This long time gap can hamper decisions about the choice and destination of seed lots [8,9].

The tetrazolium test (TZT) is a fast alternative to the germination test, aiming to assess the physiological potential of seeds and enables a detailed analysis of seeds' viability and vigor [10]. In addition to this, it also allows diagnosis of the main problems that can affect the quality of seeds in a time frame only slightly longer than 24 h [11,12], such as mechanical damages from harvest or processing, damage due to moisture or drying, damages caused by insects and other pests and deterioration during storage [13]. TZT is not affected by various conditions that can affect the standard germination test (such as

the presence of fungi) [10]. The development of this test is considered one of the main advances in seed testing in the twentieth century [6]. However, it depends on tailoring the testing parameters for each species, such as conditions for pre-soaking, concentration of tetrazolium salt solution, temperature, and conditioning duration [14]. In addition to this, for a better interpretation of the test, it is necessary to evaluate seeds one by one and to identify the embryonic structures.

The test is based on the action of dehydrogenase enzymes, which catalyze the glycolysis reactions and the Krebs cycle, components of the respiratory chain [15]. During the respiration process, these enzymes act as $H^+$ acceptors and subsequently as $H^+$ donors, i.e., the transfer of $H^+$ ions released from living tissues to the tetrazolium chloride solution, besides catalyzing the respiration process [15]. When the seed tissues absorb the clear 2,3,5-triphenyl tetrazolium chloride (TZ) solution, a stable and non-diffusible compound is formed with bright red color, called triphenyl formazan or just formazan [16]. Since the reaction occurs inside the seed cells and the compound does not diffuse, it is possible to observe the separation of colored tissues (which have cell respiration and thus are alive) and non-colored (dead) tissues [13]. The tissues undergoing deterioration take on a stronger crimson red/purple color due to the greater diffusion intensity of the TZ solution through the damaged cell membranes of those tissues [16].

This test is important for the global seed market, where companies and farmers need fast and reliable information about the quality of seed lots to make quick decisions on the marketing and planting of seeds [6]. According to [16], TZT is widely used for quality control of the seeds of many crops, forage grasses, and fruits/vegetables. For the evaluation of forest seeds, TZT has also become a promising alternative due to the same reliability and rapidity of results [9,17,18] since many forest species have seeds with dormancy, require long periods to germinate, or are recalcitrant [19]. Due to these aspects, the routine use of TZT allows for diagnosing the physiological potential more quickly and effectively, allowing faster and more accurate decisions.

Several studies have been published attesting to the viability of using this test on species of the tropical seasonally dry forest Caatinga in Brazil [20]. However, few have been focused on the genus *Handroanthus* Mattos [21,22], and to the best of our knowledge, no study has been performed on seeds of *Handroanthus spongiosus* (Rizzini) S. Grose.

*H. spongiosus* which has several popular names (cascudo, ipê-cascudo, sete-cascas, pau-d'arco, pau-d'arco-casca-fina, ipê-amarelo [23]) and is an endangered species according to Brazil's Ministry of the Environment [24]. Studies are scarce on this orthodox species reporting basic aspects, such as seed quality, multiplication, regeneration, and natural adaptation [25].

This species is commonly found in sandy soils in seasonally dry forests of the Caatinga phytogeographic domain, in intermediate stages of succession at altitudes up to 450 m [26,27]. The main use of species of the genus *Handroanthus* Mattos is by the logging industry due to the high quality of the wood [28]. This impairs the recovery of populations of the species because of the low density of adults in the reproductive phase [29]. *H. spongiosus* is frequently used together with livestock breeding in farms in the Northeast region of Brazil, mainly as a source of charcoal for household use or for brickmaking. It can also be used for the recomposition of native vegetation in degraded areas and in urban forestry because of the presence of attractive yellow flowers in October and November [26,30,31].

Due to the lack of information about the physiological potential, validation of viability protocols, and determination of the quality of this species' seeds, the aim of this study was to determine the optimal conditions for the use of the TZT to establish the viability of *Handroanthus spongiosus* seeds rapidly.

## 2. Materials and Methods

### 2.1. Plant Material

*H. spongiosus* fruits were harvested in a seasonally dry tropical forest (SDTF) area in the Pernambuco State, Brazil. Voucher specimens were deposited in the herbarium of Feira de Santana State University (HUEFS), located in the municipality of Feira de Santana, Bahia State. Two of the harvest sites were distributed in the villages of Caiçara (9°07′24.5″ S, 40°23′16.1″ W, 393 m) (HUEFS-252490) and Cristália (8°5′56.3″ S, 40°19′27.1″ W, 403 m) (HUEFS-259093), both located in the municipality of Petrolina, Pernambuco. The third harvest site was in the village of Jutaí (8°33′35.7″ S, 40°12′1.9″ W, 418 m) (HUEFS-259094), located in the municipality of Lagoa Grande, Pernambuco. All three harvest sites are in a region with semiarid climate (Köppen classification of Bsh), with average air temperatures between 25 °C and 30 °C throughout the year, and low annual rainfall, between 300 mm and 1000 mm, concentrated from November to April [32,33]. The region has high solar radiation, low relative humidity (generally lower than 50%), and high evapotranspiration (>1500 mm year$^{-1}$), resulting in negative water balances for 7 to 11 months of the year. The sparse rainfall is distributed unevenly in space and time, and the soils are shallow and crystalline [27,34].

We harvested mature fruits from 15 mother trees at each site, using a trimmer and impermeable tarp. In choosing the mother trees, we considered size, vigor, and health, as well as a minimum spacing of 20 m between trees. Harvest The fruits were collected in the district of Cristália in November 2019, while those in the villages of Caiçara and Jutaí were harvested in November 2021. To conduct the tests, we formed three lots representing the harvest sites and year. After processing, the seeds were stored in a cold chamber (adjusted to 10 ± 3 °C and 60 ± 4% relative humidity) until the start of the experiment in April 2022.

### 2.2. Water Content of the Seeds (%)

The moisture content of the seeds was determined using two subsamples of 50 seeds from each of the three lots. These seeds were placed in aluminum capsules, which in turn were placed in an oven at 105 ± 3 °C for 24 h for drying. The weight differences were expressed as a percentage of dry weight in relation to the weight of fresh seeds (adapted from 7).

### 2.3. Germination Test (G%)

We performed the standard germination test in parallel with the tetrazolium test utilizing the three lots to interpret the results of the latter test.

Since seeds of this species do not have any dormancy, there was no need for pre-germination treatments. The germination test was performed with four repetitions of 25 seeds, using three sheets of Germitest® paper moistened with a volume of water equal to the dry weight of the paper multiplied by 2.5. The seeds were placed to germinate in paper rolls, packed in plastic bags, and maintained at 25 °C in a biochemical oxygen demand (BOD) chamber with 12 h photoperiod, adapted from [7,25,35]. The germination percentage was determined after 14 days by observing at least 2 mm emission of the main root, adapted from [7].

### 2.4. Tetrazolium Test (TZT)

The experimental design was completely randomized with a 4 × 4 factorial scheme (four tetrazolium salt solution concentrations and four immersion times). We used 100 seeds, subdivided into four repetitions of 25 seeds for each treatment.

We initially evaluated the optimal conditions for the test by adjusting the procedure for removing the tegument (pre-conditioning in water), along with the concentration of solutions of tetrazolium salt (2,3,5-triphenyl tetrazolium chloride—NEON brand), temperature, and exposure time to the solution.

The seeds were placed in acrylic germination boxes (gerboxes) measuring 10 × 10 × 5 cm, each on a paper towel sheet moistened with a volume of water equal to the weight of the

dry paper multiplied by 2.5, kept on a tabletop in the laboratory (average temperature of 27 °C) for 16 h. After this pre-soaking period, the tegument was removed from each seed (Figure 1). Then the seeds were immersed for 1, 2, 3, or 4 h in tetrazolium salt solution at concentrations of 0.01%, 0.05% [36], 0.075%, and 0.1% [21], with pH adjusted to 6.5. The seeds in the solutions were then maintained in the dark in a BOD chamber adjusted to 30 °C.

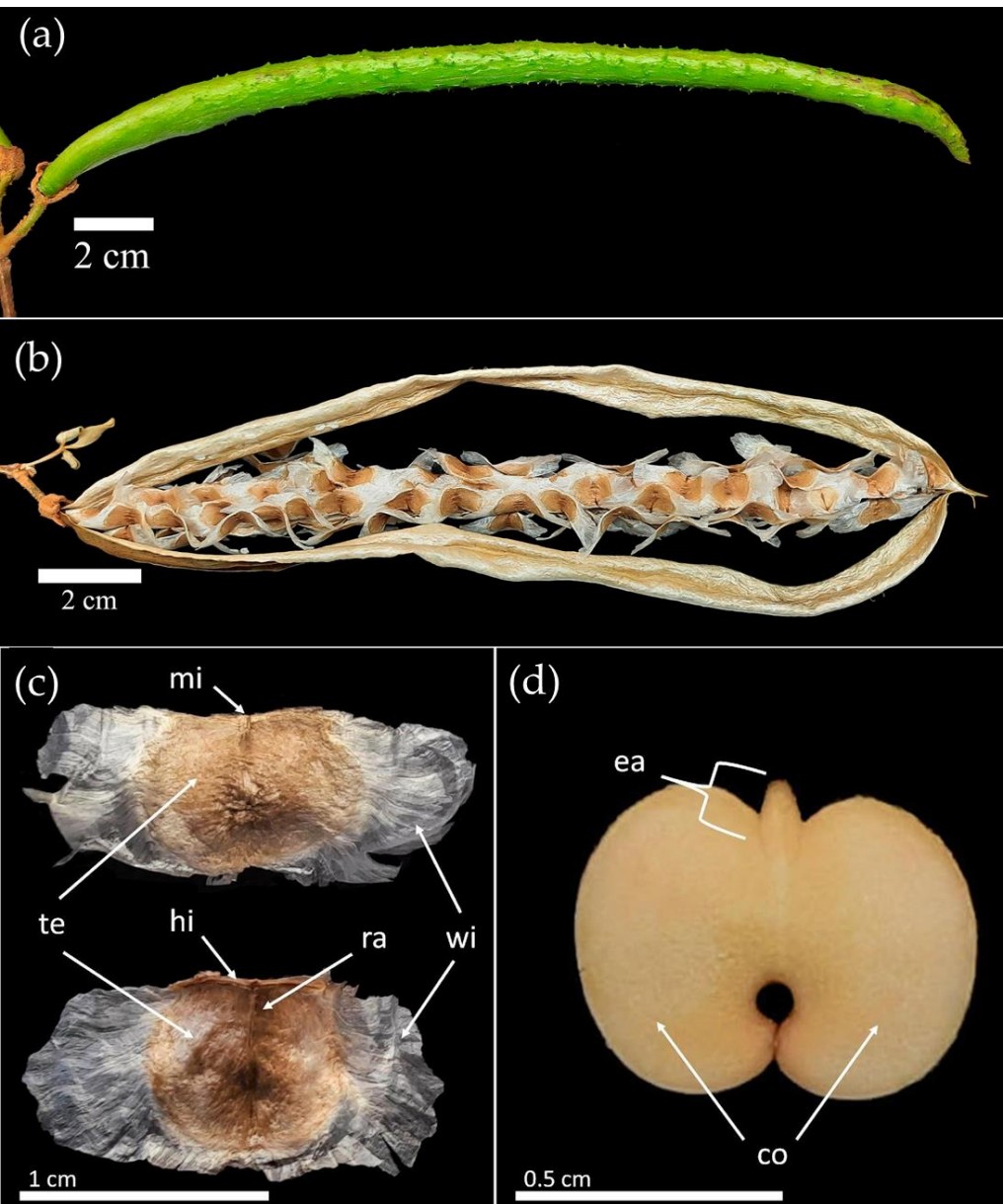

**Figure 1.** Morphology of the seeds of *Handroanthus spongiosus* (Rizzini) S. Grose (Bignoniaceae). (**a**) immature fruit; (**b**) Seeds dispersing; (**c**) seed with tegument; (**d**) seed without tegument; (mi) micropyle; (te) tegument; (hi) hilo; (ra) raphe; (wi) wings; (ea) embryonic axis; (co) cotyledons.

*2.5. Analysis of the Data*

Shapiro–Wilk test [37] was applied to evaluate the normality of the residuals of the seed viability values, while Levene test [38] was applied to determine the homogeneity of variances.

We then applied generalized linear models (GLMs) to the data because the residuals (errors) of the model failed to have normal distribution. After the analysis of the GLMs, we analyzed the significant differences in each factor (solution concentration, immersion time, and viability percentage) by pairwise comparisons with the post-hoc Tukey test at 5%

significance, with adjustment of the means by the Šidák method [39]. To ensure the best fit of the methodology, we compared the viability percentage data obtained by the TZT with the germination percentage data obtained by the Dunnett test [40]. All the analyses were carried out with the R software [41].

### 3. Results and Discussion

The newly harvested and processed seeds showed, prior to storage, water content of 5.29%, 6.90%, and 5.28%m, respectively, for Caiçara, Cristála, and Jutai lots. After storage and before the test, these were 5.24%, 5.26%, and 5.67% for the lots harvested at Jutaí, Caiçara, and Cristália, respectively. The lots were, therefore, considered uniform regarding water content, which contributed to diminishing the differences in speed during the pre-conditioning and favored the standardization at the moment of the reaction (staining) of the tissues [15]. The water content of seeds is closely associated with various aspects of their physiological quality. It can indicate the maturity stage and harvest period as well as influence the storage and possible pre-germination treatments [42]. The water content is also affected by the chemical composition and temperature of the seeds [43].

The seeds of *H. spongiosus* generally have a water content of around 5%, according to previous studies, to evaluate their physiological quality [25,44]. These dry-winged seeds are dispersed in November and December by the wind (anemochory).

The data on the percentage of viable seeds obtained by the TZT failed to present normally distributed residues ($p < 0.05$) nor homogeneity of variances ($p < 0.05$) for any seed lot (Table 1). The generalized linear models (GLMs) indicated a significant interaction ($p < 0.01$) between the studied effects for all lots.

**Table 1.** Analysis of deviance (ANODEV) for viability (%) of seeds of *Handroanthus spongiosus* (Rizzini) S. Grose (Bignoniaceae) from different lots, submitted to the tetrazolium test (TZT) with different concentrations and immersion times.

| Cristália/2019 | | | | | |
|---|---|---|---|---|---|
| Source of variation | DF | DF Difference | Deviance | Difference of Deviance | *p*-value |
| Null | | 63 | | 1294.25 | |
| Concentration (C) | 3 | 60 | 87.12 | 1207.13 | <0.01 |
| Time (T) | 3 | 57 | 984.26 | 222.88 | <0.01 |
| C × T | 9 | 48 | 133.78 | 89.10 | <0.01 |
| CV (%) = 24.12 | W ≤ 0.001 | L ≤ 0.001 | | | |
| Caiçara/2021 | | | | | |
| Source of variation | DF | DF Difference | Deviance | Difference of Deviance | *p*-value |
| Null | | 63 | | 1050.95 | |
| Concentration (C) | 3 | 60 | 56.18 | 994.77 | <0.01 |
| Time (T) | 3 | 57 | 935.87 | 58.90 | <0.01 |
| C × T | 9 | 48 | 16.74 | 42.16 | <0.01 |
| CV (%) = 22.22 | W ≤ 0.001 | L ≤ 0.001 | | | |
| Jutaí/2021 | | | | | |
| Source of variation | DF | DF Difference | Deviance | Difference of Deviance | *p*-value |
| Null | | 63 | | 935.05 | |
| Concentration (C) | 3 | 60 | 87.15 | 847.90 | <0.01 |
| Time (T) | 3 | 57 | 759.97 | 87.93 | <0.01 |
| C × T | 9 | 48 | 36.60 | 51.33 | <0.01 |
| CV (%) = 26.41 | W ≤ 0.001 | L ≤ 0.001 | | | |

Where: DF: degrees of freedom; W: Shapiro–Wilk test; L: Levene test; CV: coefficient of variation.

Freshly harvested seeds from the Caiçara, Cristália, and Jutaí lots had germination percentages of 73%, 91.5%, and 90%, respectively. Storage had little effect on seed germination, which was 72%, 88%, and 90%, respectively, for Caiçara, Jutaí (one-year storage), and Cristália (3 years storage) (Table 2). *Handroanthus spongiosus* seeds stored for 18 months at low temperatures (cold chamber adjusted to $10 \pm 3\,^\circ$C and relative humidity of $60 \pm 4\%$) presented a high germination percentage higher, with no difference in relation to fresh seeds. In addition to this, the percentage of normal seedlings and seedlings' performance (shoot and root lengths) after seed storage at the mentioned conditions were statistically equal to those parameters of fresh seeds [25]. Our results show that these seeds can be stored for up to 3 years without damage to seeds.

**Table 2.** Results of the viability (%) of seeds of *Handroanthus spongiosus* (Rizzini) S. Grose (Bignoniaceae) by the tetrazolium test (TZT) at different concentrations and immersion times in the solution.

| TZ Concentration (%) | Time (Hours) | | | |
|---|---|---|---|---|
| | 1 | 2 | 3 | 4 |
| Cristália/2019 | | | | |
| G (%) = 90 | | | | |
| 0.01 | * 0 aB | * 0 dB | 82 aA | 88 aA |
| 0.05 | * 0 aB | * 39 cB | 94 aA | 90 aA |
| 0.075 | * 0 aC | * 71 bB | 95 aA | * 79 aA |
| 0.1 | * 0 aB | 96 aA | 94 aA | * 67 aA |
| Caiçara/2021 | | | | |
| G (%) = 72 | | | | |
| 0.01 | * 0 aD | * 6 cC | 67 bB | 75 bA |
| 0.05 | * 0 aC | * 46 aB | 71 aA | 77 aA |
| 0.075 | * 0 aC | 68 bB | 73 aA | * 91 aA |
| 0.1 | * 0 aC | 75 aB | 72 aA | * 94 aA |
| Jutaí/2021 | | | | |
| G (%) = 88 | | | | |
| 0.01 | * 0 aB | * 0 bB | 84 bA | 89 bA |
| 0.05 | * 0 aC | * 44 aB | 83 bB | * 98 aA |
| 0.075 | * 0 aC | * 54 aB | 85 aA | * 97 aA |
| 0.1 | * 0 aC | 86 aB | 89 aA | * 97 aA |

Where: means followed by (*) (asterisk) indicate difference in germination percentage G (%) by the Dunnett test at 5% probability. Means followed by different lowercase letters in the column differ from each other, and means followed by different lowercase letters in the row do not differ from each by the Tukey test at 5% probability. The means were adjusted by the Šidák method. G (%): germination percentage.

Recently collected *H. spongiosus* seeds generally have rapid and uniform germination, besides not having dormancy. The emission of the radicle starts 24 h after incubation in a germinator, and germination is completed in 14 days [25,35].

By using the TZT, it was possible to determine the classes of the viable (Figure 2) and non-viable seeds (Figure 3). The viable seeds had nearly uniform crimson-red color and turgid tissues (Figure 2). Some superficial damages were observed, mainly at the edges of the cotyledons, but this did not affect the viability of these seeds (Figure 2k–n). These damages might have been caused during seed harvest, processing, or tegument removal [44].

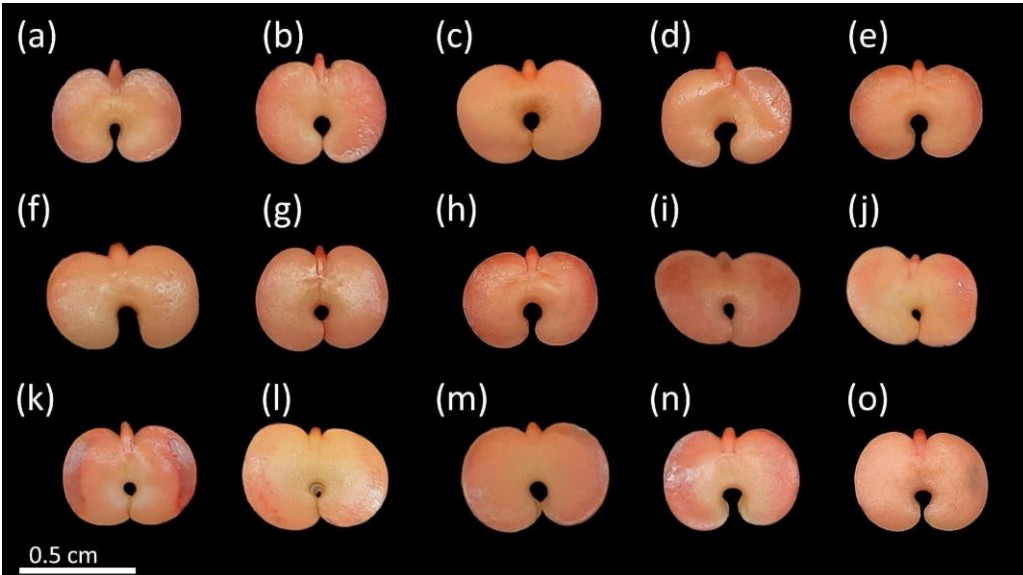

**Figure 2.** Viable seeds of *Handroanthus spongiosus* (Rizzini) S. Grose (Bignoniaceae) according to tetrazolium test (TZT). Embryo, cotyledons, and other regions of the seed with crimson red color and turgid tissues, with regions of connection between cotyledons and embryonic axis also strongly colored (**a–o**).

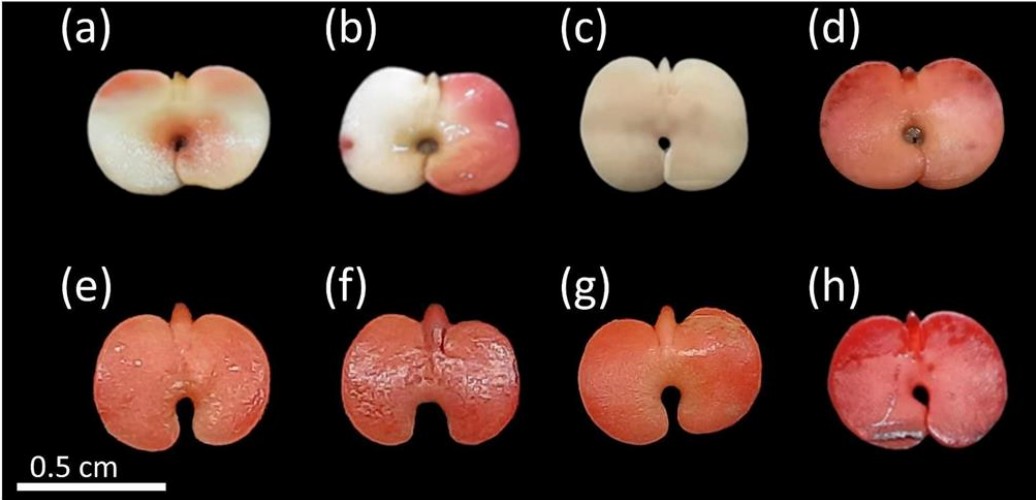

**Figure 3.** Non-viable seeds of *Handroanthus spongiosus* (Rizzini) S. Grose (Bignoniaceae) according to tetrazolium test (TZT). More than 50% of the cotyledons uncolored (**a,b**); embryonic axis and cotyledons without staining; (**c**); damaged embryo and cotyledons with intense crimson red color (**d–h**).

The seeds classified as non-viable were those with large portions of tissues without staining, mainly affecting the cotyledons and embryonic axis (Figure 3a,b), as well as the presence of structures without any staining (Figure 3c) or with damaged cell membranes (more intense color) (Figure 3d–h), also accompanied by dead tissues (without staining) at the lower end of the embryonic axis (Figure 3d,h). Figure 3e,f show tissues with flaccid texture and weak resistance to touch.

The immersion of the seeds for 1 h, at any concentration of the TZ solution, was not sufficient to reduce the TZ salt in the tissues, and there was no staining, making the distinction between viable and deteriorating tissues impossible (Table 2). Similar results were reported in *Libidibia ferrea* (Mart. ex Tul.) LP Queiroz var. ferrea seeds, in which the time of 1 h was insufficient to color the tissues [45].

The period of immersion of the seeds in the TZ solutions should be evaluated with caution. The use of solutions with low salt concentrations and insufficient exposure periods can cause underestimation of the quality indicated by the test and consequently cause mistaken interpretation of the physiological quality of the seeds [9]. The seeds immersed for 2 h in 0.05% or 0.075% tetrazolium solution showed underestimated viability results in comparison to germination percentage for all lots, while the seeds immersed in the 0.1% solution for 2 h showed viability similar to germination percentage (Table 2). Seeds immersed for 3 h at all the TZ concentrations did not present significant differences in viability percentage and germination percentage in any of the lots studied.

The seeds immersed for 4 h at concentrations of 0.075% and 0.1% showed more intense staining of all tissues (Table 2), which may hamper distinguishing living from highly damaged tissues (Figure 3). More intense staining of seeds after the TZT is associated with greater difficulty in differentiating tissues and identifying lesions, making it possible to confuse highly vigorous tissues with those having weak vigor [46–49]. The immersion of the seeds for 4 h was efficient in evaluating the viability of the seeds only at the lowest concentration tested (0.01%) (Table 2).

During the period of pre-conditioning adjustment of the seeds in distilled water, it was possible to observe the development of dark spots along the total length of the seeds, possibly due to imbibition damage when the seeds were soaked for longer than 18 h, which could have caused damages to the tissues (Figure 4). In addition to this, there was rapid absorption of water in the first 16 h. The establishment of a limit on pre-conditioning in distilled water serves to start measuring the metabolic activity and, consequently, the validation of the test with adequate staining of the tissues.

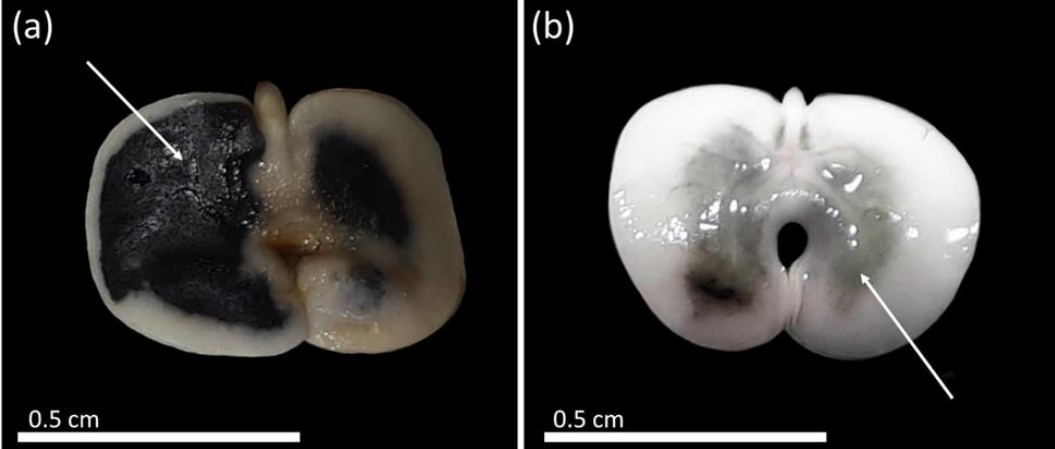

**Figure 4.** Formation of dark spots in seeds of *Handroanthus spongiosus* (Rizzini) S. Grose (Bignoniaceae) after soaking for 18 h in distilled water. (**a**,**b**) dark spots after soaking in distilled water for 18 h.

The ideal temperature for soaking of some species of the genera *Handroanthus* and *Tabebuia* can vary between 25 and 30 °C, while the soaking time varies between 12 and 24 h [9,21,36,50]. These parameters indicate the time necessary for the water to hydrate all the seed tissues, overcoming the resistance levels imposed, which are variable among species.

The preliminary test results indicated that the pre-conditioning of the *H. spongiosus* seeds in distilled water for 16 h favored the removal of the tegument and was sufficient to facilitate the gradual entry of the tetrazolium solution in the tissues, as can be noted by the effective staining (Figure 2 and Table 2).

As observed in this study, the living tissues had a light red to bright crimson color (Figure 2), in contrast to the tissues in the advanced stage of deterioration, which had an intense red color, bordering on purple, caused by the rapid diffusion of the solution through the damaged cell membranes (Figure 3d–h). In these damaged tissues, the cell membranes cannot resist the flow and diffusion of the solutions, which can easily react in

the target tissue. The dead tissues, in turn, did not have final coloring by the TZT, indicated by the presence of milky-white or yellowish-white areas and flaccid tissues (Figure 3a–c).

Some authors recommend tetrazolium solutions in concentrations ranging from 0.5% to 1.0%, with staining times ranging from 6 to 24 h for agricultural species in general [13]. Specifically for some forest species, the concentrations can vary between 0.05% and 1%, with staining times ranging from 1 to 48 h [20]. These concentrations and staining times should be adjusted for each species due to the wide variety of seed shapes and compositions.

In other studies, the greatest efficiency in detecting the viability of seeds of *ipê* trees (*Handroanthus* spp. and *Tabebuia* spp.) was obtained with immersion in 0.05% TZ solution for 4 h at 40 °C for seeds of *T. aurea* and 24 h at 36 °C for *T. roseoalba* [9,36]. The *H. spongiosus* seeds in this study needed shorter times and lower concentrations (Table 2) due to the smaller size and easier exposure of cotyledons and embryos to the solution.

In this work, the best combinations of immersion time and concentration of tetrazolium solution, which presented uniformity, visual clarity in distinguishing living and dead tissues, and coherence with the results of the germination test of the *H. spongiosus* were 2 h with 0.1% TZ solution and 3 h with any concentration, at 30 °C in the dark.

Since the intention of using the TZT is as an alternative to the use of germination and vigor tests, with faster results [5], we recommend using the shortest time and lowest solution concentration tested for time and reagent cost saving [18]. However, special attention should be paid to the immersion time to avoid underestimating the physiological quality of the seeds [9,44].

## 4. Conclusions

The comparison between the tetrazolium test and the germination test of *H. spongiosus* seeds showed that the TZT is an efficient and reliable method to assess the viability and vigor of these seeds. The intense coloring produced in the seeds by the tetrazolium solution at concentrations of 0.075% and 0.1% after 4 h hampered the differentiation of the viable tissues and can cause an overestimation of the results. Thus, the optimal conditions are removal of the tegument followed immediately by immersion in the solution of 2,3,5-triphenyl tetrazolium chloride at a concentration of 0.01% for 3 h to reduce time and costs. The use of this test can save time (two weeks) in relation to the germination test, thus providing faster results to nursery owners and seed producers. This protocol is a useful tool to monitor the viability and vigor of *H. spongiosus* seed lots for the management of programs for reforestation and conservation of this species.

**Author Contributions:** Writing, review, and Editing, J.d.J.S.; conceptualization, visualization, investigation, writing, review, editing, and supervision, C.R.P.; investigation, writing and review, R.A.G.; investigation, writing, and review, M.A.R.F.; conceptualization, visualization, investigation, writing, review, and supervision, B.F.D. All authors have read and agreed to the published version of the manuscript.

**Funding:** The authors are grateful for the assistance provided by the Brazilian Coordination for the Improvement of Higher Education Personnel (CAPES; Finance Code 001) and by the State University of Feira de Santana (UEFS).

**Institutional Review Board Statement:** Not applicable.

**Informed Consent Statement:** Not applicable.

**Data Availability Statement:** The datasets generated during and/or analyzed during the current study are not publicly available but are available from the corresponding author upon reasonable request.

**Acknowledgments:** The authors appreciate the research grants from the Foundation for the Support of Science and Technology of the State of Pernambuco (FACEPE), the Brazilian Coordination for the Improvement of Higher Education Personnel (CAPES; Finance Code 001), the State University of Feira de Santana (UEFS).

**Conflicts of Interest:** The authors declare that they have no conflict of interest or competing interests.

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
