# Peer review of "Physiological Potential of Seeds of Handroanthus spongiosus (Rizzini) S. Grose (Bignoniaceae) Determined by the Tetrazolium Test"

_2674-1024, doi:10.3390/seeds2020016_

Round 1

Reviewer 1 Report

The manuscript is well written, concise and grab the attention of the reader. The content of the manuscript may not be adequate to publish as a full paper. It would be good to include some of the preliminary data to improve the content of the manuscript. 

Really commend on the excellent figures, really a highlight of the manuscript.

Minor grammar and sentence structure errors to be corrected including some better word choices in some parts of the manuscript (see edits and suggestions)

Author Response

Dear Editor and Reviewers,

We are very grateful for your careful evaluation and your insights into our manuscript. We attempted to make the changes you recommended. We enclose here the revised manuscript "Physiological potential of seeds of Handroanthus spongiosus (Rizzini) S. Grose (Bignoniaceae) determined by the tetrazolium test" and a list of actions related to each comment presented during the review process. All the authors agreed with this revised version.

We look forward to a final evaluation of this manuscript. Please do not hesitate to contact us for further queries/changes you might require.

Thank you for your efforts on this submission.

Sincerely,

Reply to Reviewer: JHB

Line 22. Suggested to change to The seeds subjected to all treatment concentrations

Answer: Accepted

Line 24. Suggested to change to TZT treatment of 0.01%

Answer: Accepted

Line 26. Include the species name as a keyword

Answer: We did not include the species name as a keyword because it is already in the title. Thus, we chose to use the most popular name.

Line 39 e 97. “to”, “collection sites”

Answer: Correction done

Reply to Reviewer: #2

Line 20. line 20 in the Abstract.  Need to tell the reader what "BOD" means.  Another option is to delete "BOD" and simply say that seeds were incubated at 25 C.

Answer: The information has been removed.

Reply to Reviewer: JAE 21-155

Question 1. Introduction

“Please provide the more details and use value of Handroanthus spongiosus. Also provide the references about previous studies conducted on germination of this species.”

Answer: We have included the main works associated with the species, along with information on the places of occurrence of the species, types of vegetation and main uses.

Question 2. Material and Method

“Authors have collected seeds two time (2019 and 2021) and experiment was conducted in 2022. However, they didn’t mention the initial seed viability (immediately after collection). In my opinion it is important because during storage the viability might change. Please provide the initial viability and at the time of conducting the experiment”

Answer: The tetrazolium test is an alternative to the germination test to determine the viability and vigor of seeds quickly. The main objective of the work was to compare the results of the two tests. For this purpose, we needed to analyze the germination of the lots of seeds we employed, which required lots with different physiological quality, thus we utilized lots of seeds collected in different years and places. The purpose of the work was to develop a protocol or methodology (pre-soaking condition, concentration and time of immersion) to adjust the tetrazolium test so that the results would be similar to the values of the germination test and could correctly express the quality of H. spongiosus seeds. Due to this reason, the fresh seeds germination would not add any important information for our results and validation of the methodology.

Question 3. Material and Method

“Why authors selected only one constant temperature (25°C) for testing the germination. Is it the best temperature for germinating seeds of this species, if yes please provide the reference or if not than please provide the justification.”

Answer: We chose this temperature because it is the ideal temperature to germinate seeds of Handroanthus spongiosus (Rizzini) S. Grose (Bignoniaceae). As requested by the referee, we have included references to studies carried out with this species indicating that the temperature of 25 °C is best.

Question 4. Material and Method

“Also clarify whether the germination testing and viability testing time is same.”

Answer: In our methodology, the germination test for this species lasted 14 days (last line of first paragraph of page 4). On the other hand, the viability test was applied after different periods (1, 2, 3 and 4 hours) as stated in the final paragraph of page 4.

Question 5. Material and Method

“Please provide the table separately for germination results.”

Answer: Since we aimed to compare values of germination and different TZ test methodologies, we placed the G% values in table 2, together with viability results. Furthermore, it would have been pointless to assemble a table with only these three values. We also presented a comparison between the the germination and the viability test results in the text at pages 8 and 9. Thus, we believe that the best place to report the average germination values is in table 2.

Question 6. Results and Discussion

“I haven’t find sufficient discussion. This section need to be revised thoroughly. Authors can

first write the key findings and then discuss accordingly. When writing the discussion, please

put more thought process rather than just quoting other people work.”

Answer: Information has been added

Question 7. Conclusions

“Need to improve.”

Answer: Information has been added

Reviewer 2 Report

line 20 in the Abstract.  Need to tell the reader what "BOD" means.  Another option is to delete "BOD" and simply say that seeds were incubated at 25 C. 

Author Response

(The authors gave the same response as above.)

Reviewer 3 Report

Please find the attached comment Sheet. 

Author Response

(The authors gave the same response as above.)

Round 2

Reviewer 3 Report

The MS improved lot from the previous draft. However, I still prefer that authors should provide the initial seed quality data (i.e., viability and moisture content at the time of seed collection). In introduction, authors should provide some information on type and level of seed dormancy in this species or genus.

Author Response

Thank you for your review. 
